# Improvement of Left Ventricular Graft Function Using an Iron-Chelator-Supplemented Bretschneider Solution in a Canine Model of Orthotopic Heart Transplantation

**DOI:** 10.3390/ijms23137453

**Published:** 2022-07-05

**Authors:** Gábor Szabó, Sivakkanan Loganathan, Sevil Korkmaz-Icöz, Ágnes Balogh, Zoltan Papp, Paige Brlecic, Péter Hegedüs, Tamás Radovits, Matthias Karck, Béla Merkely, Gábor Veres

**Affiliations:** 1Department of Cardiac Surgery, University of Heidelberg, 69120 Heidelberg, Germany; gabor.szabo@uk-halle.de (G.S.); korkmaz_sevil@hotmail.com (S.K.-I.); paigebrlecic@yahoo.com (P.B.); radovitstamas@yahoo.com (T.R.); matthias.karck@med.uni-heidelberg.de (M.K.); gaborveres@yahoo.com (G.V.); 2Department of Cardiac Surgery, University of Halle (Saale), 06120 Halle, Germany; phegedues@gmail.com (P.H.); merkely.bela@gmail.com (B.M.); 3Division of Clinical Physiology, Department of Cardiology, Faculty of Medicine, University of Debrecen, 4032 Debrecen, Hungary; agnesbalogh@med.unideb.hu (Á.B.); pappz@med.unideb.hu (Z.P.); 4HAS-US Vascular Biology and Myocardial Pathophysiology Research Group, Hungarian Academy of Science, 1122 Budapest, Hungary; 5Heart and Vascular Center, Semmelweis University, 1122 Budapest, Hungary

**Keywords:** custodiol, orthotopic heart transplantation, iron chelator

## Abstract

Demand for organs is increasing while the number of donors remains constant. Nevertheless, not all organs are utilized due to the limited time window for heart transplantation (HTX). Therefore, we aimed to evaluate whether an iron-chelator-supplemented Bretschneider solution could protect the graft in a clinically relevant canine model of HTX with prolonged ischemic storage. HTX was performed in foxhounds. The ischemic time was standardized to 4 h, 8 h, 12 h or 16 h, depending on the experimental group. Left ventricular (LV) and vascular function were measured. Additionally, the myocardial high energy phosphate and iron content and the in-vitro myocyte force were evaluated. Iron chelator supplementation proved superior at a routine preservation time of 4 h, as well as for prolonged times of 8 h and longer. The supplementation groups recovered quickly compared to their controls. The LV function was preserved and coronary blood flow increased. This was also confirmed by in vitro myocyte force and vasorelaxation experiments. Additionally, the biochemical results showed significantly higher adenosine triphosphate content in the supplementation groups. The iron chelator LK614 played an important role in this mechanism by reducing the chelatable iron content. This study shows that an iron-chelator-supplemented Bretschneider solution effectively prevents myocardial/endothelial damage during short- as well as long-term conservation.

## 1. Introduction

Heart transplantation (HTX) is the gold standard therapy for end-stage heart failure. Presently, the one-year survival rate of recipients is 90% [1,2]. However, we are confronted with growing numbers of potential recipients without an increase in donor numbers, leading to an increasing organ deficit [1]. Various options have been suggested in the literature to resolve this issue [3]. One approach involves reducing the number of unused potential donors by extending the maximum conservation time and thereby increasing the pool of suitable recipients.

A commonly used cardioplegic conservation solution in HTX is Custodiol (Bretschneider solution, HTK-solution) [4,5]. We believe that further development of this well-established solution may improve donor organ conservation. We have previously described areas where the Custodiol solution could be improved [6,7,8]. It is common knowledge that ischemia/reperfusion during cardiac surgery and transplantation causes myocardial and endothelial injury [9,10,11]. It has also been shown that, paradoxically, the protective hypothermia used during routine cardiac surgery itself causes potentiated myocardial damage, mainly due to an increase in chelatable iron concentrations and the resulting formation of reactive oxygen species [12]. Additionally, histidine—a main component of Custodiol—has demonstrated cytotoxic effects [13]. Taking this into account, a novel amino-acid-fortified and iron-chelator-supplemented Custodiol-based solution, Custodiol-N, was developed. (Table 1) In our previous studies, we first evaluated the effect of iron chelators and N-alpha-acetyl-L-histidine on endothelial function after long-term cold storage in vitro. We found that endothelial function was preserved by our improvements [8]. Then, we studied the Custodiol-N solution’s capabilities in a rodent model of ischemia/reperfusion injury in the heart. Here, we confirmed the superiority of Custodiol-N compared to Custodiol [7]. Finally, Custodiol-N was tested in an animal model of cardiopulmonary bypass. Again, we showed that myocardial function, as well as endothelial function, could be effectively preserved by Custodiol-N [6].

However, the evaluation of the Custodiol-N solution in a clinically relevant large animal HTX model has yet to be conducted. In the present study, we focused (1) on functional recovery in comparison to the standard HTK-solution, (2) on the possible extension of safe conservation time, and (3) elucidated some particular mechanisms of action of the novel solution which were not investigated in the previous studies. Here, special attention was paid to Ca^2+^-dependent force generation and to iron homeostasis.

## 2. Results

### 2.1. Series 1

#### 2.1.1. Hemodynamic Measurements

Hemodynamic variables did not significantly differ among the groups at a baseline level. After reperfusion, compared to the baseline, mean arterial pressure (MAP) was significantly decreased in the Custodiol and Custodiol-N groups, without any significant difference between the groups. However, cardiac output (CO) was significantly lower in the Custodiol group than in the Custodiol-N group (Table 2). Additionally, in terms of ESPVR and PRSW, systolic cardiac function was significantly better preserved in the Custodiol-N group (Figure 1).

#### 2.1.2. Coronary Blood Flow

While CBF decreased significantly after transplantation compared to baseline levels in the Custodiol group, the Custodiol-N group did not show this difference. Also noticeable in a direct comparison of both experimental groups was the significant lowering of CBF in the Custodiol group (Table 2). Similar results were observed for endothelium-dependent vasorelaxation in response to acetylcholine: compared to the baseline, endothelial function was significantly reduced in the Custodiol group, while no adverse effects were seen in the Custodiol-N group (Figure 2) Meanwhile, no significant differences were observed regarding endothelium-independent vasorelaxation in response to nitroglycerin (Figure 2).

#### 2.1.3. High-Energy Phosphates

Both ATP levels and the ECP were significantly improved when using Custodiol-N, compared to the Custodiol group (Table 3).

### 2.2. Series 2

#### 2.2.1. Hemodynamic Measurements

Hemodynamic variables did not significantly differ among the groups at baseline.

After HTX, Custodiol groups showed macroscopic signs of severe ischemic injury and no satisfactory contractile function could be detected. Not a single could be weaned from heart-lung machine. Therefore, no hemodynamic measurement is available. However, in the Custodiol-N groups contractility was still preserved after 8 h and 12 h (Table 4 and Figure 3). The animals were hemodynamically stable under inotropic support. Even after 16 h cold ischemic conservation time 5 out of 9 animals could be weaned from the cardiopulmonary bypass. Two of the remaining four animals underwent massive coronary air embolism and were considered as technical failure. In two cases, the animals could not be weaned from cardiopulmonary bypass, without any evidence of technical failure. Heart rate, LV systolic pressure and CO did not significantly differ over the time in any of the Custodiol-N groups. We detected a slight decrease in MAP values in the 8 h group, which reached the level of statistical significance in the case of 12 h and 16 h conservation (Table 4). LV contractility in terms of endsystolic-pressure-volume-relationship (ESPVR) and preload-recruitable-stroke-work (PRSW) was preserved (Figure 3).

#### 2.2.2. Coronary Blood Flow

After 8 h and 12 h of ischemic conservation with Custodiol-N, CBF was significantly increased during reperfusion, compared to baseline levels. However, there were no significant results in the 16 h group (Table 4).

#### 2.2.3. In Vitro Vascular Function

As a supplementary part of series 2, with a cold ischemic preservation time of 24 h, in vitro endothelial function showed the following results. In the Custodiol group, a marked impairment of coronary endothelial function was demonstrated by a reduced maximal relaxation of isolated coronary artery segments to acetylcholine, and by a rightward shift of the concentration–response curve when compared with the native control group. Custodiol-N successfully prevented this ischemia/reperfusion-induced endothelial impairment. Endothelium-independent vasorelaxation was not altered (Figure 4).

#### 2.2.4. In Vitro Force Measurement

In the Custodiol group, it was difficult to prepare isolated cardiomyocytes for force measurements after 12 h of ischemic time, as, by then, most cells were in a state of ischemic contracture. With a sarcomere length of 1.9 µm, the calcium-sensitivity of the contractile filaments of cardiomyocytes was significantly lower in the Custodiol group after 12 h, when compared to native controls. This reduction was reversed in the Custodiol-N group (Figure 5).

#### 2.2.5. Histology

Histopathological signs of severe ischemic damage (necrosis, myofibrillar fragmentation, cellular swelling, and tissue edema) were observed in the LV myocardium of the transplanted hearts that were stored for 8 h and 12 h in Custodiol; these signs of damage were less pronounced in the Custodiol-N groups (Figure 6).

The necrosis score was significantly lower in the Custodiol-N group than in the Custodiol group both at 8 h (2.2 ± 0.4 vs. 0.3 ± 0.1, *p* < 0.05) and 12 h (2.6 ± 0.5 vs. 0.4 ± 0.1, *p* < 0.05) of preservation time.

### 2.3. Series 3

#### Determination of Chelatable Iron Concentrations

The concentrations of chelatable iron were significantly higher in the Custodiol than in the Custodiol-N group after 4 h, as well as 12 h, of ischemia. Even at 12 h, the Custodiol-N groups did not differ significantly from the native control hearts. However, hearts conserved in Custodiol for 12 h showed significantly higher levels than native control hearts. There was a tendency towards higher iron concentrations after 12 h of preservation than after 4 h of preservation for both the Custodiol and Custodiol-N groups (Figure 7).

## 3. Discussion

Even though the number of potential HTX recipients has increased by 50% since 2004 [14], the number of donors has remained almost unchanged. Nevertheless, donors are regularly excluded from heart donation because of the limited preservation time, during which the organ must reach a suitable recipient [1]. Improving the utilization rate of grafts is therefore an important goal in transplant surgery.

A wide range of over than 167 different cardioplegic and organ preservation solutions have been developed [15,16] and are presently used. Among these organ preservation solutions, the University of Wisconsin, Celsior, and Custodiol solutions are the most frequently used mixtures [17]. In terms of survival, the long-term results of these three solutions are comparable, with the Celsior solution at a slight disadvantage (one year survival: 90% for the Celsior solution vs. 92% for both the Custodiol and Wisconsin solutions) [15]. Similar results were reported in a meta-analysis by Yongnan Li et al. In this study, the Celsior solution was also associated with a significantly elevated mortality rate compared to the Wisconsin solution [18]. Aside from some sporadic comparative studies, there has been no definitive clinical trial to determine the best preservation solution. This might also explain the significant variations in global cardioplegia and organ preservation practices [17]. In Europe, the main organ preservation solution is Custodiol. In North America however, the University of Wisconsin solution takes first place. In the present study, we took a closer look at Custodiol, as it is Europe’s first-choice solution. Custodiol-N is the evolutionary next step of Custodiol, which, at its core, remains a histidine-tryptophane-ketogluterate (HTK) solution. However, it has been altered to improve its organ preservation capabilities. In the present study, as a first step, Custodiol-N was compared to its “mother solution”. In future studies, it should also be compared to other available organ preservation and cardioplegic solutions.

Another strategy for prolonging the ex vivo preservation times of donor hearts, and one that has received a lot of attention, involves ex vivo perfusion and conservation systems. Oxygenated blood is used to perfuse the graft, thereby reducing graft damage. Besides isolated case reports describing a conservation time of 8 h [Heartbeat device [19] to 10 h [OCS by TransMedics [20], reliable, clinically relevant studies are still missing. A limiting factor is that activated blood and the further release of metabolic agents from the graft lead to cardiac edema and inflammation, signs of myocardial impairment [21]. Additionally, in terms of practicability, several issues must be considered: present ex vivo perfusion systems require special equipment and the training of surgeons and technicians, which results in considerably higher costs. Therefore, this organ procurement procedure may not be feasible for all hospitals.

Based on these limitations, we aimed to improve strategies involving the well-established Custodiol solution [4,5,6,22,23,24,25]. The primary target of our work was to further reduce myocardial damage during the standard conservation period of 4 h; the secondary goal was to identify an opportunity to safely achieve prolonged organ preservation without the need for specialized staff or expensive equipment.

The comparison of Custodiol to Custodiol-N in a 4 h ischemia/reperfusion setup (series 1, Figure 1 and Figure 3) showed that Custodiol-N could also have an impact in routine cardiac surgeries with ischemic times of 4 h or less; this is indicated by a significant improvement in load-independent contractility parameters such as ESPVR and PRSW (Figure 1). Remarkably, the contractility was almost recovered to baseline levels in this setup.

The secondary goal, which was to titrate the absolute borders of Custodiol-N, also showed remarkable results. In series 2, long ischemic times of 8 h, 12 h, and 16 h were evaluated. All of the Custodiol groups had macroscopic signs of severe ischemic injury in series 2. None of the animals could be weaned from cardiopulmonary bypass, not even under inotropic support. In contrast, almost all of the animals from the Custodiol-N group could be weaned from the heart–lung machine and were hemodynamically stable under inotropic support.

Improved vascular function, and thereby improved CBF, can contribute to better cardiac performance and better graft outcomes. This improvement was already seen as a secondary finding in series 1, and it was confirmed during extended ischemic conservation periods in series 2. During coronary bypass surgery, ischaemia and reperfusion lead to endothelial injuries, as well as to contractile dysfunction and morphological injuries [26]. Although enhanced formation of reactive oxygen species (ROS) has been reported to occur in both cardiomyocytes and endothelial cells [27], leucocyte–endothelial cell interactions and the increased release of ROS from leucocytes mainly affect the endothelium, resulting in endothelial dysfunction. The damaged and dysfunctional coronary endothelium is responsible for the impaired endothelial vasodilatory function of coronary arteries, which limits CBF (as demonstrated in the present experiments) and triggers a range of problems, including platelet and leucocyte adhesion and aggregation, leading to impaired cardiac performance. In our experimental setup, Custodiol-N improved CBF. Again, to find the absolute limitations of Custodiol-N, we investigated the 24 h ischemic conservation time and found a significant impairment in the Custodiol group, but not in the Custodiol-N group.

Several mechanisms behind these protective effects of Custodiol-N have already been described in our previous studies. Some of the agents used in Custodiol have been replaced by substances that have been further developed. One of these is histidine, which, on the one hand, is an excellent buffer but, on the other, also contributes to cell toxicity [7]. It was partly replaced by N-α-acetyl-L-histidine, which shares similar capabilities but has a reduced cellular uptake [6,7,13,23]. Furthermore, the membrane-stabilizing amino acids glycine and alanine have been added to Custodiol-N [6,7,23,28]. In addition to these well-described mechanisms, we were able to explore and confirm the key mechanisms of Custodiol-N.

Myocardial dysfunction and impairment of force are known to be directly linked to sarcomere function [29,30]. Sarcomere function itself is dependent on its sensitivity to Ca^2+^, which, again, can be altered by hyper- or hypo-phosphorilation. This sensitive balance between proteinkinase A, which indirectly increases the phosphorylation of sacromeric proteins, and proteinkinase C, which has contrary effects, is influenced by ischemia/reperfusion injury [31]. As a consequence, both increased and decreased Ca^2+^ sensitivity indicate a dysregulated phosphorylation of sacromeric proteins, and, therefore, severe damage to the myocardial tissue. Our in vitro measurements of the Ca^2+^ sensitivity of myocardial sacromeres showed a significant decrease compared to native controls (Figure 5). Custodiol-N preserved this complex balanced system, and thereby preserved myocardial function.

In addition to these known mechanistic factors, we focused on the generation of reactive oxygen species (ROS) using the Fenton reaction as a key mechanism [12,23]. The Fenton reaction is boosted by chelatable iron [7,12]. To address this, two different types of iron chelators were introduced: deferoxamine and LK614 (1-(N-hydroxy-N-methylcarbamide)-3,4-dimethoxybenzol). LK614 is smaller and more lipophilic than deferoxamine, which allows membrane permeability [7,23,32]. For the first time, we proved a reduction in chelateable iron content in vivo using an exclusively developed assay (Figure 7). Together with our previous work, in which we showed a significant reduction in myocardial apoptosis [7], the present results give us insights into the link between iron chelators and the reduction in iron content, and, as a result, between apoptosis and the preservation of myocardial function.

In the past, also other pharmacologic approaches have been taken to improve pre-existing cardioplegic solutions for prolonged ischemic storage. Baxter and colleagues demonstrated the conservation of rat hearts for 16 h using the Wisconsin solution supplemented with nitroglycerine [33]. In a rat model, Kevelaitis et al. suggested the Na+/H+ exchange inhibitor cariporide, as well as the mitochondrial KATP channel (mitoKATP) agonist diazoxide, as additives to the Celsior solution for prolonged storage [34]. Similar results were seen for BMS-180448—another mitoKATP agonist—in a rat working-heart model [35]. However, none of these approaches could be advanced to a clinical trial. In contrast, our experiments were conducted in a clinically relevant canine model, and suggest a safety margin of between 12 h and 16 h. These results encouraged us to conduct a clinical trial, which is presently ongoing.

In summary, compared to Custodiol, Custodiol-N has superior protection abilities at standard ischemic times. Unlike alternative approaches, such as ex vivo organ perfusion, Custodiol-N can be used very cost efficiently. Additionally, the results are promising for extended ischemic conservation periods. The mechanisms behind these effects are linked to a stabilization of Ca^2+^ sensitivity on a sarcomeric level, and to the reduction in the chelateable iron pool. Therefore, our present approach for organ conservation may lead to improvements in cardiac surgery.

## 4. Materials and Methods

### 4.1. Animals

Foxhounds (20–35 kg; WOBE Ltd. Budapest, Hungary) were housed in a room at a constant temperature of 22 ± 2 °C with a 12-h light/dark cycle, and were fed a standard laboratory canine diet with water ad libitum. All animals received humane care in compliance with the “Principles of Laboratory Animal Care,” formulated by the National Society for Medical Research, and with the “Guide for the Care and Use of Laboratory Animals”, which was prepared by the Institute of Laboratory Animal Resources and published by the National Institutes of Health (NIH Publication No. 86–23, revised 1996). This study was approved by the appropriate institutional review committee (ethical committee of the University of Heidelberg, Medical Faculty, University of Heidelberg, Germany Ethical [Approval No 35-9185.82/A-28/06] and the Ethical Committee in Hungary [Approval No 22.1/1163/3/2010]), and is reported in accordance with ARRIVE guidelines.

### 4.2. Experimental Groups and Study Design

The study was subdivided into three series (Figure 8): 

In the first series, standard cardiac preservation times (4 h cold ischemic conservation) were utilized for an initial comparison of Custodiol and Custodiol-N. Baseline hemodynamic measurements, in vivo endothelial function, and high-energy phosphate content were compared.

In the second series, longer conservation periods were evaluated. Hemodynamic measurements and coronary blood flow (CBF) were measured after 8, 12, and 16 h of ischemic conservation time. Additionally, in the 8 h and 12 h groups, we examined histology and cardiomyocyte sensitivity to calcium. Based on the possibility of weaning some animals from cardiopulmonary bypass in the 16 h group, we examined the in vitro endothelial function of both Custodiol and Custodiol-N, compared to native control hearts, after 24 h conservation.

In the third series, the efficacy of the iron chelators LK-614 and deferoxamine, key components of Custodiol-N, were tested against Custodiol and native control hearts after 4 h and 12 h of ischemic conservation. 

### 4.3. Orthotopic Heart Transplantation

Cardioplegic arrest was induced using either the Custodiol solution or the Custodiol-N solution. Non-ischemic hearts served as controls. Donor hearts were preserved with the corresponding organ preservation solution for 4 h, 8 h, 12 h, or 16 h. Then, orthotopic HTX was performed, followed by a 2 h reperfusion period. For the measurement of iron content in series 3 (Figure 8), heart samples were taken after 10 min of reperfusion.

### 4.4. Hemodynamic Measurements

A Millar pressure-conductance catheter was used to measure pressure–volume relationships and calculate load-independent indices of myocardial contractility. CBF was evaluated using an ultrasonic flow meter placed on the left anterior descending coronary artery of the donor heart.

In series 1, endothelial function was tested in vivo: coronary endothelium-dependent and endothelium-independent vasodilatation were assessed after the administration of an intracoronary bolus of acetylcholine (10–7 M) or nitroglycerin (10–4 M), respectively.

### 4.5. In Vitro Organ Bath Functional Experiments

Organ bath experiments were performed as previously described [36]. Briefly, after the removal of the recipient animals’ hearts, the coronary arteries were isolated carefully, transversely cut into 4 mm-wide rings, immersed in cold (+4 °C) cardioplegic solution (Custodiol or Custodiol-N), and stored for 24 h before evaluation. Coronary rings of the control group were immediately mounted in the organ bath without ischemic storage. Hypochlorite (200 µM, 30 min), a highly reactive oxygen species, was added to mimic the free radical burst, which usually occurs in vivo during reperfusion. In each vessel ring, endothelium-dependent (acetylcholine) and endothelium-independent (sodium nitroprusside) vasorelaxation were investigated.

### 4.6. In Vitro Force Measurements

After the mechanical isolation of myocardial tissue samples (the original hearts of the recipient animals served as controls), permeabilization was performed with 0.5% Triton-X 100 detergent, and a single cardiomyocyte was attached to two thin stainless-steel needles, which were connected to a force transducer (SensoNor, Horten, Norway) and to an electromagnetic motor (Aurora Scientific Inc., Aurora, Canada). Isometric force measurements were performed during repetitive activation–relaxation cycles at 15 °C, first at a sarcomere length (SL) of 1.9 μm and then at an SL of 2.3 μm. Ca^2+^ contractures were evoked by transferring the myocyte from a Ca^2+^-free relaxing solution to activating solutions of gradually increasing Ca^2+^ concentrations. Active isometric and passive forces were normalized for the cardiomyocyte cross-sectional area. Isometric force values were normalized for the maximal Ca^2+^-activated active force. Then, Ca^2+^-force relations were plotted to determine the Ca^2+^-sensitivity of isometric force production, i.e., pCa50.

### 4.7. Histology

Myocardial tissue samples were taken from hearts conserved in Custodiol and Custodiol-N (8 h and 12 h), as well as from native control hearts; these samples were fixed in buffered paraformaldehyde solution (4%) and embedded in paraffin. Then, 5µm-thick sections were stained with hematoxylin/eosin for histopathological evaluation.

Necrosis was evaluated through the scoring of myocardial injuries in each group as follows: a score of 0 represented no damage. A score of 1 represented the presence of nuclear pyknosis, karyorrhexis, and karyolysis, and eosinophilic cytoplasm enhancement associated with the local infiltration of the inflammatory cells. A score of 2 indicated the absence of the natural myocardial structure and the presence of extensive inflammatory cell infiltration, and more than two separate area of necrosis but with the area of necrosis constituting <25% of the cardiac sections. Finally, a score of 3 indicated the presence of obvious necrosis, such as coagulative necrosis, more than two separate areas of necrosis, or an area of necrosis that constituted >25% of the cardiac sections.

### 4.8. Determination of High-Energy Phosphates

The analysis was performed as described previously [6]. Briefly, 1 g of heart tissue was homogenized in 10 mL 3.5% HClO_4_ and then centrifuged. Five mL supernatant was neutralized with 1 mL triethanolamin-HCL/K_2_CO_3_ solution. Adenosine triphospate (ATP) degradation was assessed with photometry using a kinetic enzyme assay containing glycerinaldehyd-3-phosphate-dehydrogenase, 3-phosphoglycerat-kinase, glycerin-3-phosphat-dehydrogenase, and triosephosphate-isomerase. The energy charge potential (ECP) was calculated as (ATP + 0.5ADP)/(ATP + ADP + AMP).

### 4.9. Determination of Total and Chelatable Iron Concentrations

To avoid a physiologically occurring quick normalization of iron concentrations, samples were taken after 10 min of reperfusion time. A new assay has been established for the determination of chelatable iron, detailed as follows:

PhenGreen calibration curve: physiological saline was incubated overnight with Chelex 100 (Bio-Rad) (5 g/100 mL) to chelate all free iron in the solution. Then, PGSK (Life technologies, Darmstadt, Germany) was diluted to 15 different concentrations between 0 and 50 µM. Dilutions were spectrophotofluorometrically analyzed at 488 nm excitation and 505–530 nm.

Sample measurements: 50 mg heart tissue was suspended in the pretreated saline at 1 mg/10 µL weight/volume ratio before homogenization and centrifugation. Supernatant was taken and incubated with ascorbic acid and then with 50 µM PhenGreen. Afterwards, samples were analyzed at 488 nm excitation and 530 nm emission.

### 4.10. Statistics

All data are expressed as the mean ± standard error of the mean (SEM). Statistical analyses of data were performed using GraphPad Prism 7.02 software (GraphPad Sofware, Inc., San Diego, CA, USA).

Series 1 and 2: individual means between the groups were compared by one-way analysis of variance, followed by an unpaired t-test with Bonferroni correction for multiple comparisons. Regarding hemodynamic and CBF measurements, the 8, 12, and 16 h Custodiol-N groups were compared to baseline results using a paired t-test, as the animals in the Custodiol groups could not be weaned from cardiopulmonary bypass. A value of *p* < 0.05 was considered statistically significant.

Series 3: ANOVA was used to compare the differences between the groups, followed by Scheffe’s post-hoc test. A value of *p* < 0.05 was considered statistically significant.

## Figures and Tables

**Figure 1 ijms-23-07453-f001:**
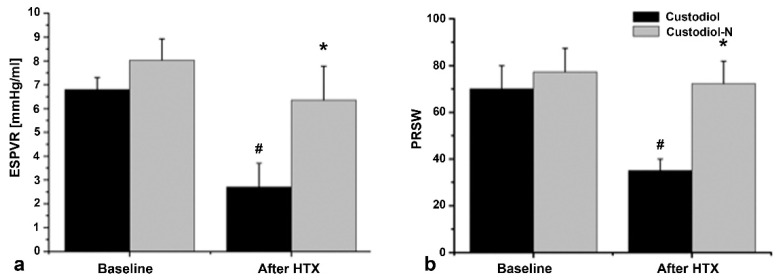
Hemodynamics (Series 1: 4 h conservation time). (**a**) End-systolic pressure–volume relationships (ESPVR); (**b**) preload recruitable stroke work (PRSW). Values given as mean ± SEM. * *p* < 0.05 vs. control, # *p* < 0.05 vs. corresponding baseline. *n* = 8 in each experimental group.

**Figure 2 ijms-23-07453-f002:**
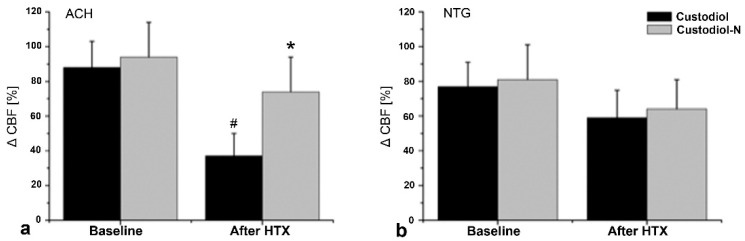
Coronary blood flow (Series 1: 4 h conservation time). Percentage change of coronary blood flow (CBF) after (**a**) acetylcholine (ACH) or (**b**) nitroglycerin (NTG). Mathematic formula: “((CBF + ACH or NTG)/(CBF before application of vasodilator))—1”. Values given as mean ± SEM. * *p* < 0.05 vs. corresponding control, # *p* < 0.05 vs. corresponding baseline. *n* = 8 in each experimental group.

**Figure 3 ijms-23-07453-f003:**
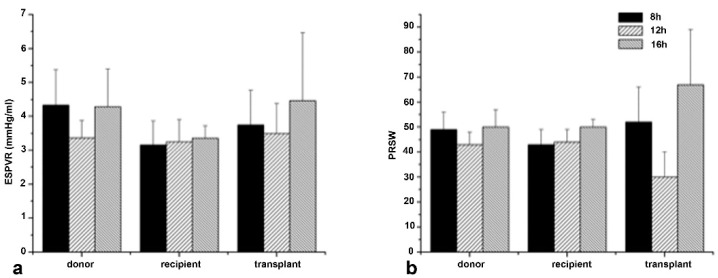
Hemodynamics (Series 2: long conservation time). (**a**): End-systolic pressure-volume relationships (ESPVR); (**b**): Preload recruitable stroke work (PRSW). *n* = 8 in each experimental group.

**Figure 4 ijms-23-07453-f004:**
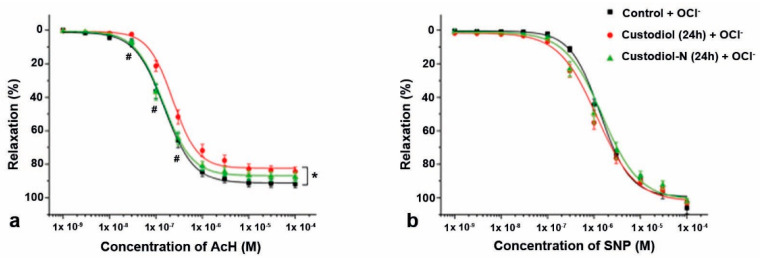
In vitro vascular function. (**a**): Acetylcholine (ACh)-induced, endothelium-dependent vasorelaxation of native control isolated coronary arterial segments, and of arteries that underwent cold ischemic conservation for 24 h in Custodiol/Custodiol-N with added reactive oxidant hypochlorite (200 µmol/L). (**b**): Sodium nitroprusside (SNP)-induced, endothelium-independent vasorelaxation of native control isolated coronary arterial segments, and of segments that underwent cold ischemic conservation for 24 h in Custodiol/Custodiol N with added reactive oxidant hypochlorite (200 µmol/L). Each point on the curves represents the mean ± SEM of 9–15 experiments in the different groups. * *p* < 0.05 vs. control, # *p* < 0.05 vs. Custodiol.

**Figure 5 ijms-23-07453-f005:**
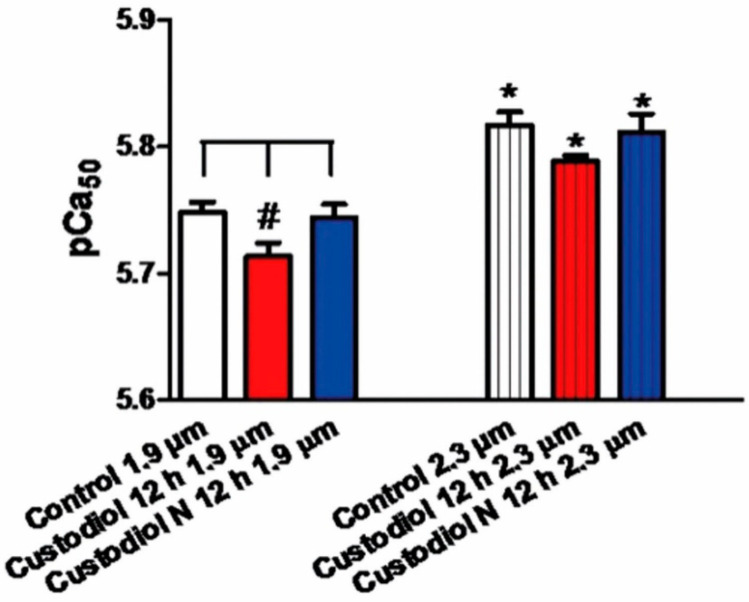
In vitro force measurements. The calcium-sensitivity index pCa50 of isolated cardiomyocytes, at 1.9 and 2.3 µm sarcomere lengths, from native control hearts and transplanted hearts from the Custodiol and Custodiol-N groups after 12 h of ischemic conservation. Columns represent the mean ± SEM of 5–17 experiments in the different groups. * *p* < 0.05 vs. 1.9 µm, # *p* < 0.05 vs. as depicted.

**Figure 6 ijms-23-07453-f006:**
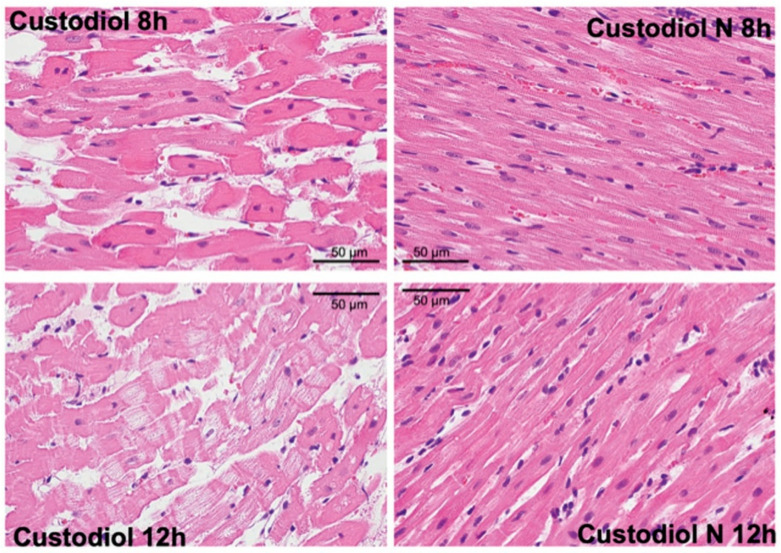
Histology. Representative histopathological sections (haematoxylin–eosin staining) of the left ventricular myocardium of transplanted hearts in each experimental group.

**Figure 7 ijms-23-07453-f007:**
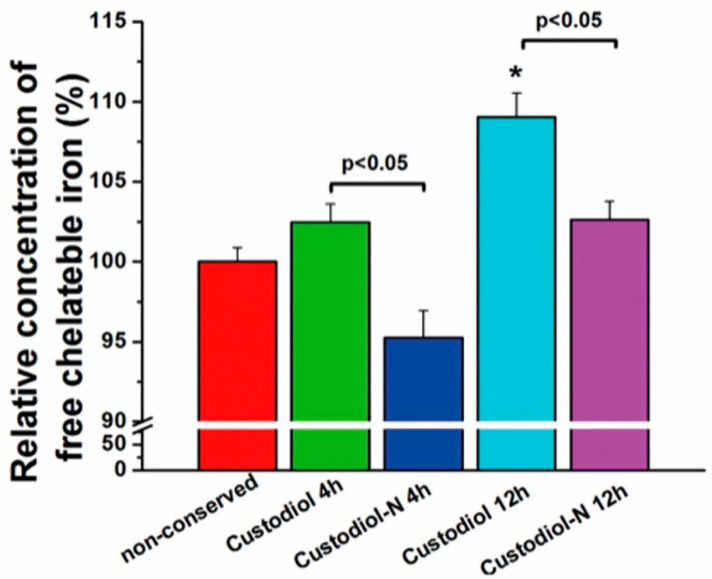
Chelatable iron concentration (Series 3). Chelatable iron concentrations are expressed as relative concentration normalized to native controls. Values given as mean ± SEM, * *p* < 0.05 vs. non-conserved.

**Figure 8 ijms-23-07453-f008:**
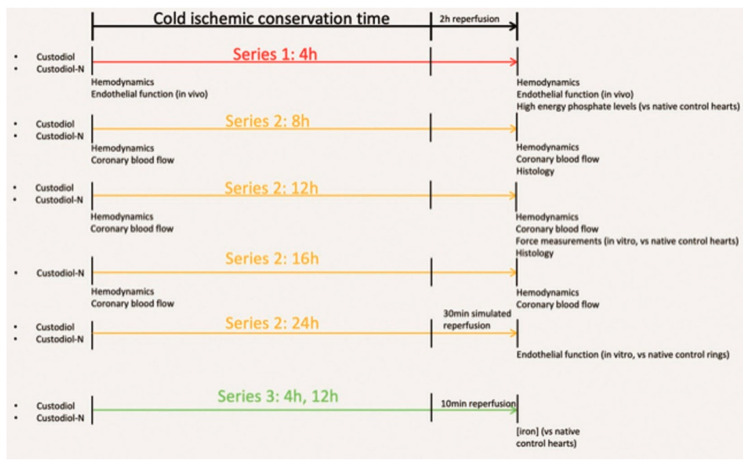
Study Design.

**Table 1 ijms-23-07453-t001:** Compounds of evaluated cardioplegic solutions.

	Custodiol (mmol/L)	Custodiol-N (mmol/L)
Na^+^	15	16
K^+^	10	10
Mg^2+^	4	8
Ca^2+^	0.015	0.020
Cl^−^	50	30
L-Histidine	198	124
N-α-Acetyl-L-Histidine	-	57
Tryptophan	2	2
A-Ketoglutarat	1	2
Aspartate	-	5
Arginine	-	3
Alanine	-	5
Glycin	-	10
Mannitol	30	-
Sucrose	-	33
Deferoxamine	-	0.025
LK-614	-	0.0075

**Table 2 ijms-23-07453-t002:** Hemodynamic variables before and 2 h after orthotopic transplantation (Series 1: 4 h conservation time).

	Donor Baseline		After Htx	
	Custodiol	Custodiol-N	Custodiol	Custodiol-N
HR [beats/min]	119 ± 8	132 ± 6	110 ± 12	135 ± 7
MAP [mmHg]	85 ± 6	90 ± 4	59 ± 6 ^#^	62 ± 2 ^#^
CO [L/min]	2.22 ± 0.11	2.53 ± 0.44	1.86 ± 0.48	2.84 ± 0.32 *
CBF [mL/min]	49 ± 5	50 ± 9	28 ± 3 ^#^	55 ± 7 *
LVSP [mmHg]	111 ± 14	107 ± 5	90 ± 7 ^#^	96 ± 6 *
LVEDP [mmHg]	7 ± 4	4 ± 2	6 ± 2	8 ± 3

HTX indicates heart transplantation, HR—heart rate, MAP—mean aortic pressure, CO—cardiac output, CBF—coronary blood flow, LVSP—left ventricular systolic pressure, values given as mean ± SEM. # *p* < 0.05 vs. corresponding baseline, * *p* < 0.05 Custodiol-N vs. Custodiol.

**Table 3 ijms-23-07453-t003:** High-energy phosphates (Series 1: 4 h conservation time).

	Before Transplantation		After Transplantation	
	Custodiol	Custodiol-N	Custodiol	Custodiol-N
ATP [µmol/g drw]	13.2 ± 1.7	14.4 ± 0.7	4.5 ± 1.1 ^#^	14.6 ± 1.7 *
ADP [µmol/g drw]	3.8 ± 1.1	5.3 ± 0.6	3.6 ± 1.3	3.8 ± 0.7
AMP [µmol/g drw]	1.9 ± 0.4	1.3 ± 0.2	2.9 ± 1.4	0.6 ± 0.1
ECP	0.78 ± 0.03	0.81 ± 0.03	0.57 ± 0.03 ^#^	0.86 ± 0.06 *

ATP—adenosine triphosphate, ADP—adenosine diphosphate, AMP—adenosine monophosphate, ECP—energy charge potential. Values given as mean ± SEM, # *p* < 0.05 vs. baseline, * *p* < 0.05 Custodiol-N vs. Custodiol.

**Table 4 ijms-23-07453-t004:** Hemodynamic variables before and 2 h after orthotopic transplantation (Series 2: long conservation time).

	8 h Conservation Custodiol-N		12 h Conservation Custodiol-N		16 h Conservation Custodiol-N	
	Donor Baseline	After Htx	Donor Baseline	After Htx	Donor Baseline	After Htx
HR [beats/min]	130 ± 10	149 ± 15	127 ± 6	156 ± 11	137 ± 7	146 ± 8
MAP [mmHg]	74 ± 4	61 ± 4	76 ± 6	58 ± 6 *	86 ± 6	60 ± 9 *
CO [L/min]	2.4 ± 0.2	2.4 ± 0.3	2.4 ± 0.3	2.2 ± 0.6	1.5 ± 0.1	1.6 ± 0.4
CBF [mL/min]	19.3 ± 2.5	39.4 ± 4.7 *	21 ± 3	40 ± 3 *	35 ± 6	43 ± 7
LVSP [mmHg]	92 ± 4	76 ± 5	94 ± 5	82 ± 7	94 ± 6	86 ± 16
LVEDP [mmHg]	8 ± 1	24 ± 4 *	9 ± 1	22 ± 5	3.1 ± 1.1	17 ± 4 *

HTX—heart transplantation, HR—heart rate, MAP—mean aortic pressure, CO—cardiac output, CBF—coronary blood flow, LVSP—left ventricular systolic pressure. Values given as mean ± SEM, * *p* < 0.05 vs. corresponding baseline.

## Data Availability

The data that support the findings of this study are available on request from the corresponding author.

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
