# Peer review of "Improvement of Left Ventricular Graft Function Using an Iron-Chelator-Supplemented Bretschneider Solution in a Canine Model of Orthotopic Heart Transplantation"

_ijms, 2022, doi:10.3390/ijms23137453_

Round 1

Reviewer 1 Report

To the authors

The reviewer appreciates the opportunity to review the present analysis by Dr. Szabó and colleagues. In the manuscript ijms-1769992 entitled Improvement of left ventricular graft function using iron chelator supplemented Bretschneider solution in a canine model of orthotopic heart transplantation, the authors investigated advantages of a cardioplegia solution, Custodiol-N, for heart transplantation (HTx) in a canine model. Added to their previous investigations on Custodiol-N, they studied the benefit in HTx compared with conventionally used Custodiol solution. They found that Custodiol-N was favorable in preventing myocardial and endothelial damages than Custodiol.

The current investigation seems quite valuable to seek for ideal cardioplegia solution in HTx because there had been few deeply comparative studies for such purposes. Although some corrections or additive discussions should be wanted, the target issue meets the interests and scopes of the journal. Critiques are described as following.

1.     While the authors focused Custodiol solution in the current research, there were some retrospective studies including meta-analyses to compare other cardioplegia solutions. There are three major preservation solutions for HTx of the University of Wisconsin solution (UW), Celsior and histidine-tryptophan-ketoglutarate solution (HTK, including Custodiol). Kristen T Carter et al. reported in a retrospective study that UW was most widely used (48%) whereas Custodiol was less (11%) (J Surg Res. 2019, PMID: 31078900). Yongnan Li et al. showed in a meta-analysis that UW solution had better clinical outcomes for heart transplantation compared with the other two organ preservation solutions (Artif Organs. 2016, PMID: 26526678). Although there had not been any definitive clinical trials which elucidated the best preservation solution as the authors descrived, it seems inappropriate to ignore those investigations. Please provide the rationale that the authors picked up Custodiol, and discuss the advantages of Custodiol-N from other solutions particularly including UW.

2.     Coronary blood flow (CBF) (Figure 2). In both cases of acetylcholine and nitroglycerin administration, it seemed strange that all the ΔCBF (even in baseline) presented less than 100% despite their CBF increasing abilities. How did the authors calculate ΔCBF? Why did those vasodilators fail to increase CBF?

3.     Why did the authors apply nitroglycerin as an endothelium independent vasodilator in the current study despite they used sodium nitroprusside in the in vitro model (Figure 4) and in previous in vivo model (Sivakkanan Loganathan et al. J Thorac Cardiovasc Surg. 2010, PMID: 19945120)?

4.     High energy phosphates (Table 3). In the comparison after transplantation, was there no statistical difference in AMP between Custodiol (2.9 ± 1.4 umol/g drw) and Custodiol-N (0.6 ± 0.1 umol/g drw)?

5.     Hemodynamic measurements (Table 4). In the comparison among 12h conservation with Custodiol-N, was there no statistical difference in LVEDP between baseline (9 ± 1 mmHg) and after Htx (22 ± 5 mmHg)?

6.     Histology. Only representative sections were presented. How about quantitative analyses with necrotic of fibrotic cross sectional area?

Author Response

Responses to the Editor and Reviewers

We would like to thank the Editor and Reviewers for the careful evaluation of our manuscript (Manuscript ID ijms-1769992) “Improvement of left ventricular graft function using iron chelator supplemented Bretschneider solution in a canine model of orthotopic heart transplantation”. Please find our point-by-point responses to your comments below.

Reviewer 1:

We want to thank Reviewer 1 for critical evaluation of our manuscript and for supporting us in the publication of the present manuscript.

Question 1: “… Please provide the rationale that the authors picked up Custodiol, and discuss the advantages of Custodiol-N from other solutions particularly including UW.”

Answer 1: As suggested by Reviewer 1 a paragraph discussing the concerns has been added to the discussion.

Changes 1: “Presently, a wide range of over than 167 different cardioplegic and organ preservation solutions have been developed[15,16] and are presently used. Among the organ preservation solution University of Wisconsin, Celsior and Custodiol are the most frequently used mixtures.[17] Long term results in terms of survival for these three solutions are comparable, with a slight disadvantage for the Celsior solution (1 year survival 90% for the Celsior solution vs. 92% for Custodiol as well as Wisconsin solution).[15] Similar results were reported in a meta-analysis by Yongnan Li et al. Also in this study Celsior solution was associated with a significantly elevated mortality rate compared to Wisconsin solution.[18] Beside some rare comparative studies there has been no definitive clinical trial to elucidate the best preservation solution. This might also explain why the global cardioplegia and organ preservation practices vary very much:[17] In Europe the main organ preservation solution is Custodiol. In North America however, University of Wisconsin takes the 1st place. In the present study we decided to have a closer look on Custodiol as Europe’s #1 solution. Custodiol-N is the evolutionary next step of Custodiol, which in its core still is a histidine-tryptophane-ketogluterate (HTK) solution. But it has been altered to improve its organ preservation capabilities. In the present study as a first step Custodiol-N was compared to its “mother solution”. Future studies have to compare it to other available organ preservation and cardioplegic solutions. “ (line 231 – 248)

Question 2: “”…CBF (Figure 3) … In both cases of acetylcholine and nitroglycerin administration, it seemed strange that all the CBP (even in baseline) presented less than 100% despite their CBR increasing abilities. How did the authors calculate CBF? Why did those vasodilators fail to increase CBF?

Answer 2: We suppose Reviewer 1 refers to figure 2 with his question. Figure 2 has to be evaluated together with table 2. In this table the absolute values for CBF are given. Based on these values the percentage change of coronary blood flow has been calculated e.g. a value of approximately 90% means an increase of coronary blood flow by 90%. That means all the results shown in Figure 2 show an increase in the CBF, there is not a decrease in CBF as stated by the reviewer.

Obviously the figure legend seems to be misleading. Therefore, according to the question of the Reviewer the figure legend has been supplemented by the mathematical formula, which was used to calculate ΔCBF.

Changes 2: “…mathematic formula: “((CBF + ACH or NTG) / (CBF before application of vasodilator)) – 1”…”(line 108 – 109)

Question 3: Why did the authors apply nitroglycerin as an endothelium independent vasodilator in the current study despite they used sodium nitroprusside in the in vitro model an in previous in vivo models?

Answer 3: According to our experience in animal studies – especially in canine studies – sodium nitroprusside had strong systemic effects – even if applied locally to the coronary arteries. To avoid these strong effects and potential adverse effects due to low systemic blood pressure, we decided to use nitroglycerin as an endothelium independent vasodilator in the present study set-up. However, we believe that the exchange of sodium nitroprusside to nitroglycerin has no impact to the key results of this study.

Question 4: “High energy phosphates (Table 3). In the comparison after transplantation, was there no statistical difference in AMP between Custodiol … and Custodiol-N …? “… (Table 4) In Comparison among 12h conservation with Custodiol-N, was there no statistical difference in LVEDP between baseline … and after Htx…?”

Answer 4: Unfortunately, the values missed to show a significance, even though as corrected noted by Reviewer 1 there is a very strong tendency.

Question 5: “Histology. Only representative sections were presented. How about quantitative analyses with fibrotic cross sectional area?”

Answer 6: We agree to the reviewer that additional presentation of quantitative analysis could offer new insights to the reader. Therefore we have now added unpresented data of the histological analysis to the revised version of the manuscript.

Changes 6: “…Necrosis was evaluated through the scoring of myocardial injury in each group as follows: a score of 0 represented no damage; score of 1: the presence of nuclear pyknosis, karyorrhexis and karyolysis, and eosinophilic cytoplasm enhancement associated with the local infiltration of the inflammatory cells; score of 2: the absence of the myocardial natural structure and the presence of extensive inflammatory cell infiltration and more than two separate area of necrosis but the area of necrosis <25% of the cardiac sections; and score of 3: the presence of obvious necrosis such as coagulative necrosis, more than two separate area of necrosis or the area of necrosis >25% of the cardiac sections...” (line 422 – 429)

“…Necrosis score was significantly lower in the Custodiol-N group in comparison to the Custodiol group both at 8h (2.2 +/- 0.4 vs. 0.3 +/- 0.1, p<0.05) and 12h (2.6 +/- 0.5 vs. 0.4 +/- 0.1, p<0.05) preservation time…” (line 198 – 200)

Reviewer 2 Report

Very interesting study with promisinig results. Presented study had a relatively small group and provided data was obtained  from single-center. Furthermore data are collected on basis of animal samples. Despite limitations of this study, presented manuscript might be appropriate for the Journal.

Please add some details about  clinical perspective. What this study bring to everyday clinical practise? What is cost-effectivness of this new solution? When is it possible to use it in human heart transplantation?

Author Response

Responses to the Editor and Reviewers

We would like to thank the Editor and Reviewers for the careful evaluation of our manuscript (Manuscript ID ijms-1769992) “Improvement of left ventricular graft function using iron chelator supplemented Bretschneider solution in a canine model of orthotopic heart transplantation”. Please find our point-by-point responses to your comments below.

Reviewer 2:

We want to thank Reviewer 2 for careful revision of our manuscript and declaring the results as promising.

Question 1: “… Presented study had a relatively small group and provided data was obtained from single-center. Furthermore data are collected on the basis of animal samples. Despite limitations of this study, presented manuscript might be appropriate for the Journal. Please add some details about clinical perspective. What this study bring to everyday clinical practice?

Answer 1: Even though this experimental study was performed in a heart transplant setup, Custodiol-N might not only be advantageous in this setting, but also might offer benefits in routine cardiac surgery and everyday clinical practice. Routine cardiac surgical interventions have developed to be safe. Nevertheless, emergency interventions in patients with acute myocardial infaction or patents with marginal cardiac reserve still have a high mortality and mobility.

Custodiol-N has shown in this study as well as our previous studies a strong evidence to have stronger effects in the protection of cardiac tissue and preservation of myocardial function. Therefore, especially this patient collective in everyday clinical practice could benefit from Custodiol-N.

Additionally, in demanding situations where surgeons have to deal with great time pressure, such as multiple valve replacements and the previously mentioned heart transplantations, where the surgical time is highly limited, Custodiol-N can show its advantages.

Question 2: “What is the cost-effectiveness of this new solution?”

Answer 2: Even though the exact costs of Custodiol-N are not clear, it will be significantly lower than alternative such as organ conservation systems (OCS). Unlikely competing OCS devices, there will also be no training necessary for surgeons. This will also significantly reduce the costs. Custodiol-N can replace the presently widely used Custodiol solution, without adaptation of personal or instruments used in the operation theater.

Question 3: “When is it possible to use it in human heart transplantation?”

Answer 3: This question cannot be clearly answered because first of all clinical trials have to show the effectivity of Custodiol-N in routine cardiac interventions. Afterwards additional clinical trials have to be performed in high risk patients (e.g. patients with acute myocardial infarction or marginal cardiac reserve or patients having a high risk for prolonged surgical procedures). We suggest clinical trials in heart transplantation after successful completion of these studies.

Round 2

Reviewer 1 Report

The reviewer thanks the authors’ effort to thoroughly revise the manuscript along with the reviewer’s comments. All the concerns were solved to be enough satisfied, and the reviewer believes that the manuscript comes up to be more qualified.